# Adolescents’ Lifestyle Determinants in Relation to Their Nutritional Status during COVID-19 Pandemic Distance Learning in the North-Western Part of Romania—A Cross-Sectional Study

**DOI:** 10.3390/children10060922

**Published:** 2023-05-24

**Authors:** Bogdana Adriana Nasui, Rodica Ana Ungur, Gabriel Adrian Nasui, Codruta Alina Popescu, Ana Maria Hofer, Sebastian Dersidan, Monica Popa, Horatiu Silaghi, Cristina Alina Silaghi

**Affiliations:** 1Department of Community Health, “Iuliu Hatieganu” University of Medicine and Pharmacy, 6 Louis Pasteur Street, 400349 Cluj-Napoca, Romania; adriana.nasui@umfcluj.ro (B.A.N.); hoferana2000@gmail.com (A.M.H.); sebastian_dersidan@yahoo.com (S.D.); monica.popa@umfcluj.ro (M.P.); 2Department of Medical Specialties, “Iuliu Hațieganu” University of Medicine and Pharmacy, 8 Victor Babeș Street, 400012 Cluj-Napoca, Romania; ungurmed@yahoo.com; 3Law Faculty, Dimitrie Cantemir University, Teodor Mihali Street, No 60, 400591 Cluj-Napoca, Romania; nasuigabriel@yahoo.com; 4Department of Practical Abilities—Human Sciences, “Iuliu Hatieganu” University of Medicine and Pharmacy, 6 Louis Pasteur Street, 400349 Cluj-Napoca, Romania; 5Department of Surgery V, “Iuliu Hatieganu” University of Medicine and Pharmacy, Victor Babes Street 8, 400012 Cluj-Napoca, Romania; horatiu.silaghi@umfcluj.ro; 6Department of Endocrinology, “Iuliu Hatieganu” University of Medicine and Pharmacy, Victor Babes Street 8, 400012 Cluj-Napoca, Romania; alina.silaghi@umfcluj.ro

**Keywords:** schoolchildren, lifestyle, diet, remote learning, COVID-19 pandemic

## Abstract

Several studies have examined how the lockdown restrictions enforced to halt the spread of COVID-19 have affected adults’ movement behaviors; however, there is limited information regarding the effects on youth. This study aimed to report on the lifestyle habits of adolescents during COVID-19 pandemic remote learning and explore associations between the participants’ bodyweight and lifestyle behaviors. We used a cross-sectional study applied to 285 students studying in the gymnasium. The response rate was 74.21%. They completed an online questionnaire regarding lifestyle, eating habits, and nutritional status (assessed by the Body Mass Index—BMI). The study was conducted in January–February 2021. The percentage of overweight and obese was higher among boys (*p* = 0.001). The mean sleep duration was 8.12 (SD:1.284) hours per night, and was higher in boys than in girls. The respondents met the physical activity (PA) recommendation for their age, based mainly on unorganized PA. The screen time was 4–6 h or more for the majority of the respondents. Dietary habits included a high consumption of comfort food, like cereals, sweets, soft drinks, snacks, and fried food, but a lower consumption of vegetables and fast food. The regression analyses showed that the BMI was correlated with the BMI of the parents and the number of hours spent in front of the computers in free time. The study identified behavioral and environmental factors that can be modified with effective strategies to reduce overweight and obesity among school students and to promote a healthy lifestyle.

## 1. Introduction

Since December 2019, the SARS-CoV-2 virus, responsible for the COVID-19 disease, has spread across the globe and caused a global pandemic. The World Health Organization (WHO) declared it a public health emergency of international concern on 30 January 2020 [1].

On 12 March 2020, Romania declared a state of emergency [2]. The COVID-19 pandemic has had an adverse influence on the whole of society. Many national governments introduced measures to counteract the disease and avoid infections, such as social distancing policies, closures of schools, shops and leisure opportunities, and contact restrictions. This unprecedented situation led to significant changes in children’s daily routine; they no longer attended school and out-of-school activities (e.g., participation in sports, free play at playgrounds, etc.), but were isolated at home with their families. Furthermore, self-isolation has also been linked to boredom and stress, which lead to higher consumption of energy-dense comfort food and emotional eating. Home quarantine and online schooling have led to unhealthy food alternatives, increased time spent in front of screens, chaotic sleeping hours, and it being challenging to practice physical activities (PA) outside the home. Typically, children undertake many additional activities besides school, such as horse riding, swimming, dancing, and learning foreign languages. The limitation of these activities also limited contact with friends, which significantly increased efforts to maintain relationships through social media and various types of communicators [3]. This significantly influenced their health, including nutrition, physical exercise, sleep, and social functioning, increasing their media use [4].

Any unhealthy eating habits in the growth period of life can lead to irreversible consequences such as increased risk of obesity, non-communicable diseases, and decreased immune system function [5,6].

A reduction in PA during the pandemic could have harmful effects, as PA during youth is an essential determinant for future PA [7,8], preventing future obesity and cardiovascular diseases and providing mental health [7].

Restrictive social interactions imposed by the pandemic aggravated the overuse of digital devices for socializing, which included virtual dates, virtual tourism, and virtual parties and family conferences. While mindful (and regulated) use of digital devices is linked with well-being, excessive screen time is reportedly associated with a range of adverse mental health outcomes such as psychological problems, low emotional stability, and greater risk for depression or anxiety [8]. Studies from the literature showed an increase in total screen time among children, adults, and adolescents [9].

Children who regularly get the right amount of sleep benefit from better attention, behavior, learning ability, memory, and overall better mental health. Sleep deprivation can lead to high blood pressure, obesity, and depression [10]. Sleep disturbance can significantly impact the ability to concentrate, emotional health, immune function, and academic performance. Its deficiency increases the risk of cardiovascular disease in children and adolescents and can result in mood swings, which may be exacerbated by poor mental health caused by the COVID-19 pandemic [11]. Data from the literature regarding sleep among children and adolescents during the pandemic are controversial. Some studies reported an increase in sleep time per day during the pandemic compared with the pre-pandemic period [12,13]. Another study mentions that 40% of Chinese teenagers experienced problems falling asleep, staying asleep, and not sleeping too much [14]. 

In Romania, there is a gap with respect to studies investigating lifestyle and behaviors among adolescents during the COVID-19 pandemic, and particularly during the period of remote learning. Therefore, this study aimed to evaluate the lifestyle during remote schooling for secondary school students from the Northwest Romanian region (the equivalent of NUTS 2 regions in the European Union). Another aim was to assess the effect of lifestyle behaviors on the nutritional status of adolescents during the COVID-19 pandemic. We evaluated body mass index, nutrition, PA, interaction with electronic devices, and sleep.

To the best of our knowledge, this is the first study in Romania to investigate behavioral factors related to weight status during the distance learning of adolescents.

## 2. Materials and Methods

### 2.1. Study Design and Participants

We used a cross-sectional study design. The sample consisted of 285 respondents, secondary school students aged 11–14 years. In Romania, lower secondary education or gymnasium (ISCED2) includes grades 5–8 (primary education includes grades 1–4, and upper secondary education includes high school grades 9–12). The study was based on a questionnaire distributed online through Google Forms with the help of teachers and the research team. We distributed the link to the questionnaire through the Microsoft Teams platform. Selected adolescents voluntarily and anonymously completed the questionnaires in the presence of their parents in January–February 2021 when the Romanian students attended the school online (during the second wave of the pandemic). Informed consent from the parents was obtained by filling in the online questionnaire.

Data were collected from schools in the north-western part of the country. We selected the schools in which students had appropriate technology and digital tools and agreed to participate in the study. The questionnaire was created using the Google Forms tool and was distributed electronically via networks of teachers working in secondary schools in Transylvania County, Romania and personal networks. The study was approved by the Territorial Romanian School Inspectorate no. 25.01/2021.

We used a multistage cluster method to draw a representative sample of 285 adolescents from the northwestern part of the country. We divided the north-western part of Romania into six counties. In the first stage, we divided the counties into smaller urban and rural administrative areas (clusters). In the second stage, from these clusters, we randomly selected two schools (sub-clusters). The gymnasium schools (a block) were selected in the third stage. From a total of 175 gymnasium schools (based on the data from the education management information system of the Centre of Information and Technologies in Education), we generated six random numbers from 1 to 175. We selected six schools (about 2160 school students), of which three agreed to participate. From these three schools, from each grade we randomly selected a class. From the first school, from a total of 336 students we selected 120 students (35.71%), from the second school, from a total of 360 adolescents we selected 136 (37.7%), and from the third school, from 348 students we selected 128 students (36.78%) (Figure 1). Of these, we randomly selected and approached a total of 384 adolescents and parents.

We calculated the sample size, considering a response rate of 75% (based on previous studies [15,16] with a confidence interval of 95% and 5% margin of error based on the total population of gymnasium school students from the north-west region of Romania; this amounted of 94,772, according to Romanian statistical reports. We calculated a computed minimum size of 288 participants. However, 384 participants were invited to participate in the present study, based on the expected participation rate. The response rate was 74.21%. We obtained a representative sample of adolescents and secondary school students from northwestern Romania [17].

Inclusion criteria: gymnasium students, the participation of one of the selected schools, and the parent’s acceptance of their children to participate in the study. Exclusion criteria: incomplete questionnaires, absent adolescents (and parents) at the time of data collection.

### 2.2. Instruments and Variables

To achieve the proposed objectives, we used a validated structured questionnaire. The questionnaire used in the study included four parts: demographic data, dietary behaviors, PA assessment, time spent in front of computers or other devices and sleep assessment. The food frequency questionnaire was a valid, modified version adapted to Romanian habits [18,19]. The questionnaire was pretested on a sample of 30 respondents. Spearman’s correlation coefficient was used to assess the reliability (r = 0.773). The time required to fill in the questionnaire was approximately 25–35 min. Cronbach’s α coefficient was used to determine the internal consistency of the questionnaire. The value of Cronbach’s α for our questionnaire came out to be 0.86, which suggests a good internal consistency.

The questionnaire evaluated:Demographic data (age, sex, urban/rural environment, parents’ education).The students’ Body Mass Index (BMI), using self-reported weight and height. We calculated the BMI = Weight (kg)/Height^2^ (m^2^). We divided the sample into underweight, normal weight, overweight, and obese according to World Health Organization (WHO) cut-off points [20].In the case of the parents of students, we used the same formula of BMI and divided the sample into classes of obesity for adults according to WHO.Dietary assessment. We estimated the frequency of intake of the main food groups in the previous month: meat, milk and dairy, vegetables and fruits, cereals (bread, rice, pasta, etc.) and potatoes, as well as sweets intake (e.g., cakes, ice-cream, biscuits, cookies etc.), intake of sweetened beverages such as soft drinks (SSB), fast food intake, and intake of chips, fried food, and energy drinks. To assess food intake, we used the following frequency: never, less than 1/week, 1–3 times/week, 4–6 times/week, daily or more. Vegetables and fruits were assessed using the frequencies: none, 1–2/day, 3 or more portions. A portion of fruit/vegetables was considered 80 g of medium-sized fruit, 1/2 cup of chopped/cooked vegetables, 30 g of dried fruit, or 150 mL of fruit or vegetable juice. We also assessed the number of meals per day.The questionnaire estimated the level of PA. PA was assessed as frequency per week and duration. We assessed PA during online classes at the location of distance learning (e.g., at home, exercises in front of the computers), organized PA (e.g., classes of swimming and other sports), and unorganized PA (e.g., playing in the park with other children).The use of electronic devices—screen time (during remote school and in free time).Sleep as quantity (number of hours of sleep per night) and quality (if they wake up rested in the morning, bedtime).

### 2.3. Statistical Analyses

All statistical analyses were performed using SPSS for Windows, version 26.

The participant characteristics were described as means (standard deviations) and proportions, as appropriate. The quantitative variables were computed as means and standard deviations. The *t*-test was used to compare the quantitative variables. Frequencies and percentages were computed for the categorical variable. The chi-square test was used to test the association between categorical variables. The normality of distribution was assessed using Kolmogorov–Smirnov test of normality.

The association between nutritious status of the adolescents (estimated by BMI, the dependent variable) and the BMI of the parents, PA and screen time (independent variables) was performed by multiple linear regression analysis.

The results were statistically significant with *p* < 0.05.

## 3. Results

### 3.1. Demographic Characteristics of the Study Group

Out of the total of 285 secondary school students in this study, 167 were female, and 118 were male. A total of 81.48% (*n* = 132) female and 86.21% (*n* = 100) male students came from the urban environment (*p* = 0.296). The average age for girls was 12.36 ± 0.869, years and for boys, it was 12.63 ± 0.943 years. The average age of the group was 12.47 ± 2.262 years (*p* = 0.968).

The study revealed that the majority of the respondents’ parents had completed higher education studies. A total of 59.6% (*n* = 170) of mothers possessed a university degree, whereas 35.8% (*n* = 102) had high school degrees and 4.6% (*n* = 13) had only elementary studies. In comparison, 52.6% (*n* = 150) of fathers had university degrees, 41.8% (*n* = 119) high school degrees, and 5.6% (*n* = 16) had completed elementary studies.

### 3.2. Assessment of Nutrition Status%

We calculated the BMI of the adolescents and stratified the sample depending on the classes of obesity. The study results showed that the majority of the female students had a normal weight, 73.7% (*n* = 123), and only 21.6% were overweight or obese. Regarding the boys, almost half had a normal weight, 44.9% (*n* = 53), and over half, 52.5%, corresponded to the overweight or obese class (*p* = 0.001) (Figure 2).

The study evaluated the nutrition status of the parents. According to the results, the majority of the respondents’ mothers (55.4%, *n* = 158) had a normal weight and only 29.8% (*n* = 85) were overweight. Unlike the mothers, the majority of the adolescents’ fathers were overweight (49.4%, *n* = 140) or obese (29.8%, *n* = 85) (Table 1).

### 3.3. Dietary Behaviors

When studying the diet, we investigated the number of meals consumed per day, and the frequency of intake of different food groups, including but not limited to snacks fast food, fruit and vegetables.

We found that the average number of main meals consumed by girls was 3.02 (SD:0.764), and by boys was 3.01 (SD:0.536). The group mean was 3.02 (SD:0.677) (*p* = 0.759).

The results of the study revealed that a considerable percentage of the girls (22.8%, *n* = 38) and only 13.6% (*n* = 16) of boys did not eat breakfast. However, lunch was consumed by almost all pupils. In addition, a considerable number of respondents (19.87% of girls vs. 19.4% of boys) had dinner after 8:00 p.m. (Table 2).

When investigating the number of snacks consumed by the Romanian adolescents, the results showed that girls had 2.96 ± 1.54 snacks per day and boys had 2.36 ± 1.01 snacks per day (*p* = 0.129), with the mean of the sample being 2.55 ± 1.38 snacks. Snacking after dinner was common among both male and female respondents (56.3%, *n* = 94 of girls vs. 47.41%, *n* = 55 of boys), *p* > 0.05.

The analysis of fruit and vegetable intake revealed that the consumption of fruit was very popular, with 88% of respondents (both male and female) eating at least one fruit per day. However, a low vegetable intake was observed, with only 11% of boys and 19.2% of girls consuming at least three servings of vegetables per day (Table 3).

Regarding intake of other food groups, the results of the study showed a frequent consumption of cereal-based foods and potatoes, red meat, milk and acidophilic products and breakfast cereals. Fried food (meat, potatoes, fish) was also consumed with a higher frequency, and 22% of girls and 22% of boys ingested fried food four to six times per week (Table 4).

The consumption of energy drinks, chips, sweets, soft drinks and fast food among Romanian adolescents during the pandemic were considered unhealthy dietary behaviors. Energy drinks and fast food were consumed with a very low frequency. However, the intake of sweets and chips was comparatively higher. Although most respondents never consumed energy drinks (85.6% of girls and 89% of boys), sweets were very popular among the studied group, with 39.6% of girls and 33.9% of boys eating sweets at least four times per week. In addition, soft drink consumption (1–3 times/week) was reported by approximately a third of respondents (Table 5).

### 3.4. Physical Activity

The present study investigated whether the respondents were engaged in at least five hours per day of PA. The majority of the adolescents, 61.7% (*n* = 103) of the girls and 55.9% (*n* = 66) of boys in the studied sample, reported that they performed PA for at least five hours per week. There were no statistical differences regarding PA (at least five hours per week) between the boys and the girls (*p* = 0.321).

We analyzed the PA of the sample performed during distance learning in terms of organized PA and unorganized PA, frequency per week, and duration. The results showed that adolescents were engaged mostly in unorganized PA (e.g., playing outdoors) and less during remote school or organized PA. However, the total PA of the adolescents met the recommendations for a healthy lifestyle (Table 6).

### 3.5. Hours Spent in Front of the Computer (Screen Time)

When analyzing the time spent in front of the computer during distance learning, the results evidenced that the majority of the students, 49.7% (*n* = 83) of girls and 62.7% (*n* = 74) of boys, reported that they spent between 4 and 6 h on electronic devices during school hours (*p* = 0.029). Another considerable percentage of adolescents spent more than six hours during remote learning (Table 7).

In addition, the results of the study revealed that most of the girls 52.7% and a great percentage of the boys 46.6% spent more than 4 h in front of such devices during their free time (*p* = 0.582) (Figure 3).

Regarding parental control programs used on electronic devices, 32.3% (*n* = 54) of the girls and 33.1% (*n* = 39) of the boys had installed these programs (Table 8).

### 3.6. Sleep

The average number of hours of sleep slept per night by girls was 8.01 (SD:1.278), and among boys this was 8.41 (±1.258), with statistical differences between the girls and boys (*p* = 0.026) (Figure 4). The mean hours of sleep in the studied group was 8.18 (SD: 1.284).

Estimating the quality of sleep during remote schooling, our study evidenced that over half of the girls, 55.1% (*n* = 92), mentioned the fact that they had a restful sleep. In contrast, a higher percentage of the boys, 78.0% (*n* = 92), reported restful sleep, with statistical significance between the girls and the boys (*p* < 0.001) (Table 9).

We investigated the bedtime hours during remote learning. Most of the girls, 69.5% (*n* = 116), and 72.0% (*n* = 85) of the boys went bed in the evening between 10:00 and 12:00 p.m. (*p* = 0.308) (Table 10).

### 3.7. Factors Associated with Nutrition Status

We ran a multiple linear regression to predict the nutrition status of the adolescents during the pandemic period depending on gender, mother’s BMI, father’s BMI, number of hours in front of the computer, and PA for at least 5 h per week. According to our results, gender, mother’s BMI, father’s BMI, and number of hours in front of the computer were statistically significantly associated with the BMI of the respondents (*p* < 0.05) (Table 11).

## 4. Discussion

The study aimed to evaluate the lifestyle and the effect on the nutritional status of secondary school students and adolescents from the north-western part of Romania during the COVID-19 pandemic distance learning. We evaluated body mass index, nutrition, PA, interaction with electronic devices, and sleep.

The nutritional status was assessed in our study by the BMI indicator, using self-reported weight and height. According to the results of our research, over half of the boys were overweight or obese, compared to only 21.6% of girls that were overweight or obese. Data from the literature investigating BMI changes during the pandemic showed an increase in annual BMI during the COVID-19 pandemic compared to previous years, especially among children in the obese range [21]. School closures may have affected children’s increased weight gain [22]. Other studies showed a reduced pattern of PA, increased sedentary activities, and sleeping and screen time, leading to an increase in BMI [23]. In the present study, the results showed that the adolescents’ BMI was associated with the BMI of the parents and the number of hours spent on screen time in their free time during the pandemic. These results support the role of environmental factors as modifiable factors influencing the nutritional status of children.

Regarding food intake, the study’s results revealed an appropriate consumption of fruits by the majority of adolescents, 88% of girls, and 88.1% of boys consumed 1–2 or at least three portions per day. Vegetable intake was very low in comparison to the recommended intake. Most respondents did not consume three portions per day, many of them avoiding consumption altogether. These results are consistent with studies on different Romanian populational groups, confirming the trend of insufficient vegetable consumption in contrast with an appropriate fruit intake [15,24,25].

In this study, we considered comfort foods, the consumption of sweets, fast food, chips, sweetened sugar beverages, and even cereals (pasta, rice, bread, and other cereals). The present study revealed high consumption of sweets, SSB and cereals among Romanian secondary school students during the pandemic. Our results are similar to studies among children from Poland showed increased intake of sweets and reduced SSB and fast food [26]. These findings are also consistent with a survey taken in Italy among children and adolescents that evidenced an increase in sweetened snack packages [27], bread, pizza, and bakery products.

Other review studies showed an increase in snacks and sweets consumption similar to the present study [28,29]. Moreover, previous Romanian studies carried out on adolescents evidenced a high intake of SSB (especially among boys) and other types of sweets (chocolate, added sugar, candies, and cookies) [30].

Regarding the distribution of meals, we found statistically significant differences between girls and boys who did not eat breakfast (22.85 vs. 13.6%). These results are similar to other studies in the literature that revealed that Romanian pupils tended to skip breakfast [31,32]. Previous research has evidenced that childhood obesity is associated with lifestyle factors, such as lack of sleep, excess screen time, skipping breakfast, and a lower frequency of family meals [33,34,35,36].

One of the disadvantages of the present study was the lack of the data from pre-pandemic period. Previous reports from the National Institute of Public Health in our country [37] reported an obesogenic environment, with consumption based on refined cereals, added sugars and fats, and a reduced intake of vegetables, fruits and fish. This pattern of energy-dense food consumption can lead to overweight and obesity. It is possible that the results of our study evidenced dietary cultural habits of the Romanian population that were preserved and perhaps augmented during the pandemic.

Another important lifestyle component that was estimated in our study was PA. It is well established that PA and exercise are essential to improving/maintaining physical and mental health and improving quality of life [38]. Studies from the literature have evidenced that measures restricting the spread of COVID-19 led to a decline in PA for most children and adolescents, especially among boys and older children and adolescents [7]. Data from the literature evidenced that girls usually performed less PA than boys [39].

Even when students learn at home, health and physical educators should provide guidance and activities to help them meet the recommended 60 min or more of moderate-to-vigorous PA daily for children and adolescents ages 6–17 [40].

According to the results of our study, the respondents met the recommendation of daily PA, mainly due to unorganized PA (e.g., playing with other children). PA was lower, especially in boys, but the results were not statistically different. These results are similar to those of other studies from the literature concluding that children and adolescents performed less PA and more sedentary activities, and the most common PA during the pandemic was free-play/unstructured PA (e.g., going for a walk, running) [41,42].

The move to online schooling during the pandemic led school-aged children to spend many hours a day in front of the screen interacting with their teachers and classmates [43]. The results of our study showed that almost all the adolescents spent 4–6 h or more during the remote school, exceeding the recommendation of the American Academy of Pediatrics (AAP) for recreational screen time, which should be limited to less than 2 h per day for children 11–13 years old [44].

In addition, a previous study found that children’s screen time (e.g., watching television and playing video games) increased significantly from 2.6 h to 5.9 h/day (on average 3.2 h/day) during the pandemic-related school closures [45]. Evidence from the literature shows that changes in children’s screen time use are associated with greater parental involvement, a modifiable risk factor in promoting children’s well-being [46,47]. In our study, almost half of the secondary school students, both girls and boys, spent 4–6 h or more in front of different devices. Parental control was reported in a low percentage, about a third of the children (32.3% for girls and 33.1% for boys).

One other lifestyle factor that could have been modified during the pandemic and was important to consider was sleep. Sleep and mental well-being are intimately linked. Studies conducted during the pandemic suggest that COVID-19 impacted sleep and anxiety in multiple countries and across age groups [48,49]. Sleep disturbance among children was attributed to bedtimes and wakeup times, inability to do outdoor activities during the lockdown, remote learning, and lack of interpersonal social activities. Although intrinsic factors regulate sleep, extrinsic factors are essential in determining sleep duration, timing, and quality. Behavioral factors (e.g., screen time, PA, bedtime routine), physical and environmental factors (e.g., light and noise exposure), and parental factors (e.g., parental child attachment, parental education, parental stress) all affect the sleep health of children [50]. The recommended sleep amount for school-age children of 6–13 years of The Sleep Foundation Organization is about 9–11 h per night [51,52]. In our study, Romanian pupils had, on average, 8.11 hours of sleep, and significantly more for boys than for girls. The study results revealed that most respondents went to bed between 22:00 and 24:00, and a significant percentage after 24:00 (19.2% of girls and 13.6% of boys). Sleep quality was also affected; 44.9% of girls and 22.0% of boys (*p* = 0.001) reported unrestful sleep. These results are consistent with other studies that showed poor quality sleep among children and adolescents during the pandemic [53] and changes in sleep habits [48,54].

Our study is subject to some limitations. First is the method of the online questionnaire. Bias can appear through overestimation or underestimation of food intake and PA. Self-reported weight and height were other limitations of the study. Respondents with overweight, especially girls, may under-report their weight. Overweight status may be also a predictor of reporting error in height. However voluntary participation in the study may limit these unreliable answers. Objective methods to measure weight, height, and PA would lead to more accurate data results. Another limitation of our study was that we selected children from environments with digital resources to make the data collection possible. Because of voluntary participation, there may have been participation bias favoring students who are more willing to answer questionnaires. Another limitation of our study was the lack of data from the pre-pandemic period, which would have allowed us to make comparisons with the pandemic lifestyle. However, to the best of our knowledge, this is the first study from Romania that evaluates the lifestyle of adolescents during remote learning. Follow-up studies are necessary to investigate whether these behaviors will be adopted in the future.

This future research must be applied on a larger sample size to better understand the environmental factors involved in children’s lifestyles that can lead to overweight and obesity during or outside the pandemic.

These findings identify targets for behavioral and environmental interventions to reduce childhood obesity risks and promote a healthy lifestyle.

## 5. Conclusions

This study reported unhealthy behaviors in secondary school adolescents during remote learning during the COVID-19 pandemic in Romania. Our study evidenced the characteristics of lifestyle factors that were associated with the Romanian adolescents’ weight status during pandemic remote learning, namely excessive screen time and the body mass index of the parents. The present study can serve as a reference point for future studies among Romanian schoolchildren.

The results of the present study can be the basis for the implementation of public health preventive programs among children and parents to improve lifestyle behaviors, such as appropriate food choices and PA, reduced screen time, and appropriate sleep habits, to prevent weight gain and to promote the intellectual performance of secondary school students.

## Figures and Tables

**Figure 1 children-10-00922-f001:**
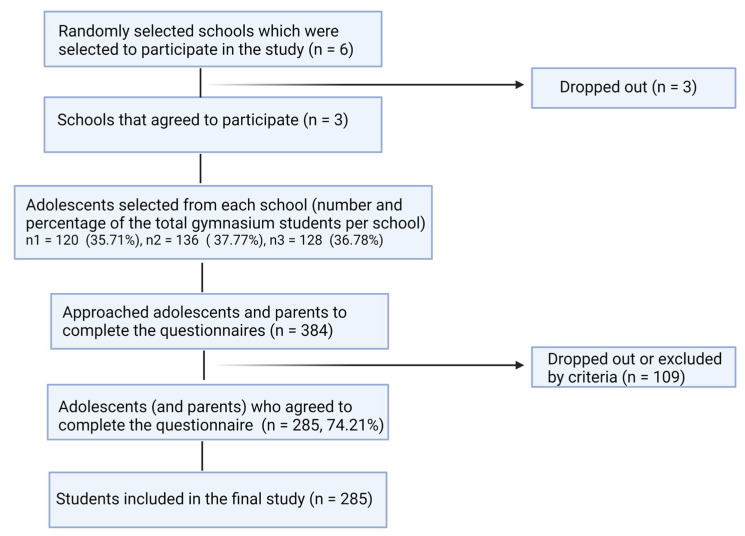
Reporting flow diagram of sample selection. Created with BioRender.com.

**Figure 2 children-10-00922-f002:**
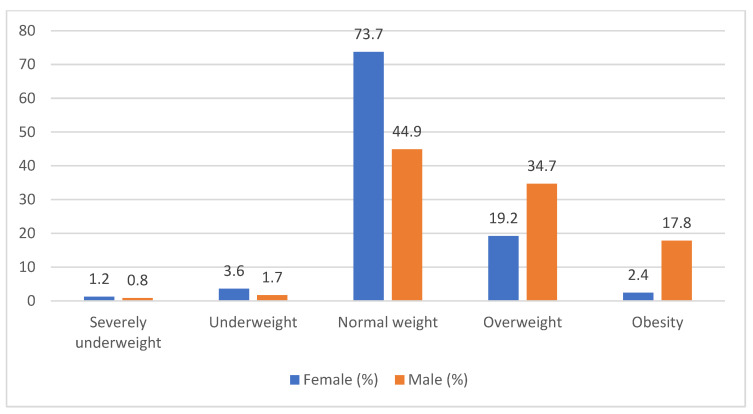
Sex distribution of BMI in the sample size.

**Figure 3 children-10-00922-f003:**
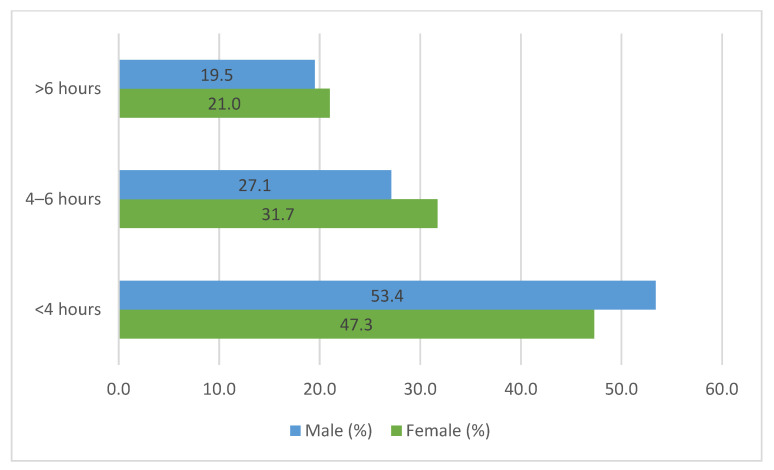
Hours spent in front of the computer (free time).

**Figure 4 children-10-00922-f004:**
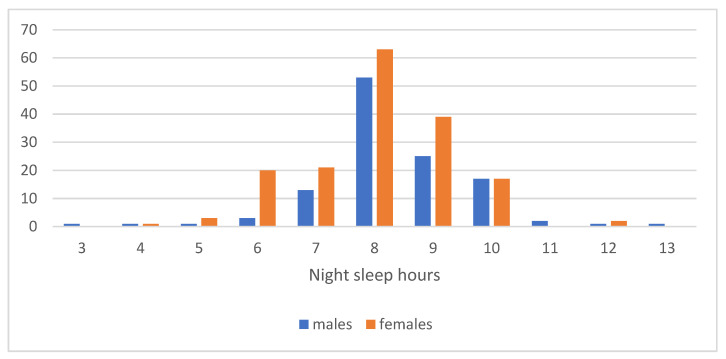
Number of hours slept per night during the pandemic.

**Table 1 children-10-00922-t001:** Body mass index for parents of children included in the study.

Variable BMI	Mothers *n* (%)	Fathers *n* (%)
Underweight	18 (6.3)	0 (0.0)
Normal weight	158 (55.4)	60 (21.1)
Overweight	85 (29.8)	140 (49.1)
Obesity	24 (8.4)	85 (29.8)

**Table 2 children-10-00922-t002:** Adolescents having breakfast, lunch and dinner during remote schooling.

Breakfast	Girls *n* (%)	Boys *n* (%)	*p* Value
Yes	129 (77.2)	102 (86.4)	0.051
No	38 (22.8)	16 (13.6)	
**Lunch**	**Girls *n* (%)**	**Boys *n* (%)**	
Yes	157 (94.0)	114 (96.6)	0.317
No	10 (6.0)	4 (3.4)	
**Dinner hour**	**Girls *n* (%)**	**Boys *n* (%)**	
<19	17 (14.4)	17 (14.4)	0.967
18–20	107 (64.1)	78 (66.1)	
20–22	31 (18.6)	22 (18.6)	
>22	2 (1.2)	1 (0.8)	

**Table 3 children-10-00922-t003:** Adolescents’ vegetable and fruit portion intake during pandemic remote schooling.

Vegetables	Girls *n* (%)	Boys *n* (%)	*p* Value
Raw	77 (46.1)	53 (44.9)	0.842
Cooked	90 (54.4)	65 (55.1)	
**Fruit**	**Girls *n* (%)**	**Boys *n* (%)**	
None	20 (12.0)	14 (11.9)	0.947
1–2/day	96 (57.5)	70 (59.3)	
3 or more	51 (30.5)	34 (28.8)	
**Vegetables**	**Girls *n* (%)**	**Boys *n* (%)**	
None	36 (21.6)	31 (26.3)	0.158
1–2/day	99 (59.3)	74 (62.7)	
3 or more	32 (19.2)	13 (11.0)	

**Table 4 children-10-00922-t004:** Intake of other food groups during remote learning.

Cereals and Potatoes Intake	Girls *n* (%)	Boys *n*(%)	*p* Value *
Never	0 (0.0)	3 (2.5)	0.171
Less than 1/week	27 (16.2)	13 (11.0)	
1–3/week	76 (45.5)	53 (44.9)	
4–6/week	33 (19.8)	29 (24.6)	
Daily or more	31 (18.6)	20 (16.9)	
**Red Meat**	**Girls *n* (%)**	**Boys *n* (%)**	***p* Value ***
Never	12 (7.2)	9 (7.6)	0.180
Less than 1/week	41 (24.6)	23 (19.5)	
1–3/week	82 (49.1)	60 (50.8)	
4–6/week	20 (12.0)	23 (19.5)	
Daily or more	12 (7.2)	3 (2.5)	
**Milk and acidophilic products**	**Girls *n* (%)**	**Boys *n* (%)**	***p* Value ***
Never	8 (4.8)	8 (6.8)	0.900
Less than 1/week	31 (18.6)	18 (15.3)	
1–3/week	71 (42.5)	50 (42.4)	
4–6/week	37 (22)	26 (22)	
Daily or more	20 (12)	16 (13.6)	
**Breakfast cereals**	**Girls *n* (%)**	**Boys *n* (%)**	***p* Value ***
Never	29 (17.4)	15 (12.7)	0.754
Less than 1/week	46 (27.5)	35 (29.7)	
1–3/week	52 (31.1)	42 (35.6)	
4–6/week	30 (18)	21 (17.8)	
Daily or more	10 (6)	5 (4.20)	
**Fried food**	**Girls *n* (%)**	**Boys *n* (%)**	***p* Value ***
Never	2 (1.2)	3 (2.5)	0.846
Less than 1/week	65 (38.9)	41 (34.7)	
1–3/week	78 (46.7)	58 (49.2)	
4–6/week	18 (10.8)	12 (10.2)	
Daily or more	4 (2.4)	4 (3.4)	

* *p* < 0.05 was considered statistically significant; chi-square.

**Table 5 children-10-00922-t005:** Unhealthy behavior Children’s consumption of energy drinks, chips and sweets during pandemic remote learning.

Energy Drinks	Girls *n* (%)	Boys *n* (%)	*p* Value *
Never	143 (85.6)	105 (89.0)	0.904
Less than 1/week	15 (9.0)	7 (5.9)	
1–3/week	5 (3.0)	3 (2.5)	
4–6/week	1 (0.6)	1 (0.8)	
Daily or more	3 (1.8)	2 (1.7)	
**Chips**	**Girls *n* (%)**	**Boys *n* (%)**	***p* Value ***
Never	19 (11.4)	19 (16.1)	0.123
Less than 1/week	86 (51.5)	72 (61.0)	
1–3/week	50 (29.9)	20 (16.9)	
4–6/week	8 (4.8)	5 (4.2)	
Daily or more	4 (2.4)	2 (1.7)	
**Sweets**	**Girls *n* (%)**	**Boys *n* (%)**	***p* Value ***
Never	5 (3.0)	2 (1.7)	0.489
Less than 1/week	28 (16.8)	29 (24.6)	
1–3/week	68 (40.7)	47 (39.8)	
4–6/week	33 (19.8)	22 (18.6)	
Daily or more	33 (19.8)	18 (15.3)	
**Fast Food**	**Girls *n* (%)**	**Boys *n* (%)**	***p* Value ***
Never	20 (12.0)	15 (12.7)	0.443
Less than 1/week	111 (66.5)	81 (68.6)	
1–3/week	23 (13.8)	19 (16.1)	
4–6/week	8 (4.8)	2 (1.7)	
Daily or more	5 (3.0)	1 (0.8)	
**Soft drinks**	**Girls *n* (%)**	**Boys *n* (%)**	***p* Value ***
Never	23 (13.8)	7 (5.9)	0.207
Less than 1/week	62 (37.1)	55 (46.6)	
1–3/week	58 (34.7)	40 (33.9)	
4–6/week	17 (10.2)	10 (8.5)	
Daily or more	7 (4.2)	6 (5.1)	

** p* < 0.05 was considered statistically significant; chi-square.

**Table 6 children-10-00922-t006:** Physical activity of the studied sample.

Duration of PA min/week	Girls	Boys	*p* Value
During online school	142.82 ± 231.68	135.85 ± 192.07	0.669
Organized	79.95 ± 184.89	97.10 ± 152.01	0.254
Unorganized PA	486.19 ± 2024.44	278.93 ± 428.05	0.152
Total	708.96 ± 2078.64	511.88 ± 541.57	0.154

Data presented as means ± SD; *t*-test.

**Table 7 children-10-00922-t007:** Hours spent in front of the computer during pandemic remote learning.

Hours Spent in Front of the Computer (Online Schooling)	Girls *n* (%)	Boysn (%)	*p* Value
<4 h	7 (4.2)	4 (3.4)	
4–6 h	83 (49.7)	74 (62.7)	0.029
>6 h	77 (46.1)	40 (33.9)	

**Table 8 children-10-00922-t008:** Parental control of screen time during pandemic remote schooling.

Parental Control	Girls *n* (%)	Boys *n* (%)	*p* Value *
Yes	54 (32.3)	39 (33.1)	0.899
No	113 (67.7)	79 (66.9)	

* *p* < 0.05 was considered statistically significant; chi-square.

**Table 9 children-10-00922-t009:** Restful sleep time during remote learning.

Restful Sleep	Girls *n* (%)	Boys *n* (%)	*p* Value *
Yes	92 (55.1)	92 (78.0)	0.001
No	75 (44.9)	26 (22.0)	

* *p* < 0.05 was considered statistically significant; chi-square.

**Table 10 children-10-00922-t010:** Bedtime hour during remote learning.

Bedtime Hour	Girls *n* (%)	Boys *n* (%)	*p* Value *
Before 22:00	18 (10.8)	17 (14.4)	0.308
Between 22:00 and 24:00	116 (69.5)	85 (72.0)	
After 24:00	33 (19.8)	16 (13.6)	

* *p* < 0.05 was considered statistically significant; chi-square.

**Table 11 children-10-00922-t011:** Multivariate associations between parents’ BMI, computer time, physical activity level and BMI of adolescents.

Coefficients						
Model	Unstandardized Coefficients		Standardized Coefficients	t	95% CI Lower Bound	95% CI Upper Bound
	B	Std Error	Beta			
Constant	9.021	1.576		5.723 *	5.623	12.059
BMI mother	0.208	0.049	0.0247	4.229 *	0.113	0.307
BMI father	0.158	0.046	0.199	3.414 *	0.065	0.260
Hours Spent in Front of the Computer (free time)	0.716	0.237	0.167	3.023 *	0.250	1.184
5 h/week of PA	−0.229	0.0378	−0.033	−0.606	−0.939	0.555

* *p* < 0.05 was considered statistically significant; CI—confidence interval.

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
