# Peer review of "Adolescents’ Lifestyle Determinants in Relation to Their Nutritional Status during COVID-19 Pandemic Distance Learning in the North-Western Part of Romania—A Cross-Sectional Study"

_children, 2023, doi:10.3390/children10060922_

Round 1

Reviewer 1 Report

To the Authors

Summary

Summary

This cross-sectional study examined associations between BMI and lifestyle habits of 285 secondary school students residing in Romania during the COVID-19 pandemic. According to the investigators, overweight/obesity and sleep duration were higher in adolescent boys than girls. Overall, screen time was ≥4-6 hours/day. Children had an unhealthy dietary pattern consisting of a high intake of cereals, sweets, soft drinks, snacks, and fried food; low intake of vegetables and fast food. Multivariate analyses revealed that children’s BMI was positively associated with parent’s BMI and hours of computer-time. The authors concluded that strategies targeting modification of these factors might effectively reduce obesity during adolescent years and promote a healthy lifestyle.

Indeed, the COVID epidemic has created great burden to everyday family life. Restrictions, social-distancing and remote-learning have brought about lifestyle changes affecting children of all ages. Understanding how these modifications influence children’s dietary habits and weight will help in the development of strategies to halt the rise in childhood obesity, thus improving children’s quality of life and future. 

The topic of this manuscript is of interest and adds new evidence to the existing database. Overall, this manuscript was well written. Nevertheless, to improve study transparency, there are minor issues that need to be accounted for.

Please refer to my comments to the authors below:

Introduction: The background and rationale for this study was well-written and supported by the literature.

Methods

-Line 128 ‘We calculated the sample size, considering a response rate of 75 % (based on previous studies)

Please add a reference for the study that the sample size was based on.

Statistical analysis

-Line 185 ‘’The participant characteristics were described as the mean (standard deviation) and proportions as appropriate.’

Please correct ‘means and standard deviations’

-Line 189 ‘The association between nutritious status of the adolescents and the BMI of the parents  ….was performed by regression analysis’

Indicate which regression analysis was applied e.g linear or logistic, the variable used as the dependent (i,e BMI score or categories) and independent variables. How the effect size was measured (ORs, β- coefficient, 95%CI) and adjustment for confounding factors.

-Table 4 typographical error , ‘Red meat’ not ‘Red meet’

Regarding presentation of data as counts and percentages, this should be indicated as n(%) not No(%). Please revise in all tables.

Table 6.

Please indicate below Table 6 that data is presented as means ± SD as well as the statistical test used to estimate p-values

-Line 289 ‘The average number of hours of sleep slept per night by girls was 8.01 (±1.278),’ can be written as ‘The average number of hours of sleep slept per night by girls was 8.01 (SD: 1.278)’

-Table 11.

In the introduction, it is mentioned that the aim of this study is to determine associations between lifestyle factors and BMI of adolescents.

Perhaps a more accurate description of table 11 would be

‘Multivariate associations between parents’ BMI, computer time, physical activity level and BMI of adolescents’

-Table 11 presents multivariate associations.

Was a univariate regression analysis conducted for each independent variable?

Please present this table before the multivariate regression Table 11 or include it as an online supplement

Discussion:

The findings were critically appraised supported by previous studies.

-Line 393-395 ‘Sleep and mental well-being are intimately linked. Studies conducted during the pandemic suggest that COVID-19 impacted sleep and anxiety in multiple countries and across age groups’.

Please add references for these studies.

Limitations:

Add the limitation of cross-sectional study design

For spelling errors please refer to the comments to the authors above.

Author Response

Dear Esteemed Reviewer

Please see the attachment of our answers. 

Kind regards

Reviewer 2 Report

The original article entitled "Adolescents' Lifestyle Determinants in Relation to Their Nutritional Status during COVID-19 Pandemic Distance Learning in the North-Western Part of Romania" (children-2387391) aimed to assess the lifestyle of secondary school students from the Northwest Romanian region during remote schooling and to evaluate their nutritional status based on behavioral and environmental factors during the COVID-19 pandemic in Romania.

Comments: In the abstract, the number of students who were offered the study and the number who actually participated should be mentioned. The standard deviation should be included when reporting the average hours of sleep. Introduction: The justification for the study is well presented with appropriate references. However, the objective is not clearly stated and is presented in three different ways. It should be rewritten to clearly state the objective of the study. The methodology is a cross-sectional design with 285 participants, which is a very small sample size. Additionally, voluntary participation may introduce selection bias towards individuals with better health. What was the participation rate? Was a sample size calculation performed with an expected prevalence? A figure illustrating the sample selection process, including the number of children in the six schools that were offered the opportunity to participate and the three that actually participated, would be helpful. This design only allows for description of the situation and does not permit analysis. The title should be adjusted to reflect a descriptive study. The sampling method is a bit confusing as it is both voluntary and done by multistage cluster method. Please clarify. A validated questionnaire was used, so the reference should be included. In the statistical analysis, the quantitative variable should be discussed, as well as how the normality of its distribution was assessed. The anthropometric evaluation of the children during the pandemic period is not solely a result of the pandemic. How can this be justified? The discussion does not evaluate pre-pandemic characteristics, so it is unclear if these characteristics are typical of the students or a result of the pandemic. Strengths and weaknesses of the study should be presented in the discussion.

Author Response

Dear Esteemed Reviewer,

Please find attached our answers to your comments.

Kind regards

Reviewer 3 Report

First of all, I would like to thank for the opportunity to revise the manuscript entitled:

„Adolescents’ Lifestyle Determinants in Relation to Their Nutritional Status during Covid-19 Pandemic Distance Learning in the North-Western Part of Romania”.

This is a well-written article that identifies an important issue in the field of science about relations between weight status and lifestyle during Covid-19 pandemic at adolescents.

The title is correct and explains the aim of the current study well.

The abstract is well written, it summarizes the article.

The introduction is well-written, the examined indicators are well-summarized.
The information about the expansion of overweight and obese children is alarming end every cause has to be assessed.

Tables and figures are well-formatted and informative, but 11 tables are too much for this article. Please  merge it, and some significations you can mark with sign  „*” in figures. It would be much more effective for the readers to analyse the manuscript.  

The discussion is well-supported by several articles, but its like an enumeration, please connect the different parts. It would be much more understandable. Please write the limitations about self-reported weight and height.

Author Response

Dear Esteemed Reviewer,

Please see attached the answers to your comments.

Kind regards

Round 2

Reviewer 2 Report

After carefully reviewing the manuscript in its second version entitled "Adolescents' Lifestyle Determinants in Relation to Their Nutritional Status during COVID-19 Pandemic Distance Learning in the North-Western Part of Romania" (children-2387391), as well as the authors' clarifications.

Comments: If a random sampling is done using clustering, I understand that each element of the cluster is each of the schools. Therefore, it should be indicated how many students there are in each school and how many students have participated in each school.

I don't understand how the sample size has been calculated. For calculating the sample size of a prevalence, which corresponds to a cross-sectional design, you need to indicate the estimated prevalence in your sample size calculation. No estimated prevalence is mentioned in the calculation.

Regarding the methodology by which you invite students to participate, it is not clear whether they receive a specific invitation individually to each of the selected students or if you expect participation from those who voluntarily wish to participate. In the latter case, there would be a participation bias, as usually the students who participate are more conscious about a healthier lifestyle. Therefore, the flowchart should provide information with the aforementioned data.

Please represent p-values with three decimal places, and revise the tables accordingly.

Table 11 should include confidence intervals.

In the conclusions, you cannot claim evidence for lifestyle modifications since you do not have previous data.

I believe this study allows identifying the general characteristics of adolescents in Romania during the pandemic and could serve as a reference point for future studies.

Author Response

Dear Esteemed Reviewer,

Kind regards
